# Clinical Characteristics of Osimertinib Responder in Non-Small Cell Lung Cancer Patients with EGFR-T790M Mutation

**DOI:** 10.3390/cancers11030365

**Published:** 2019-03-15

**Authors:** Akihiro Yoshimura, Tadaaki Yamada, Naoko Okura, Takayuki Takeda, Kazuki Hirose, Yutaka Kubota, Shinsuke Shiotsu, Osamu Hiranuma, Yusuke Chihara, Nobuyo Tamiya, Yoshiko Kaneko, Junji Uchino, Koichi Takayama

**Affiliations:** 1Department of Pulmonary Medicine, Graduate School of Medical Science, Kyoto Prefectural University of Medicine, 465 Kajii-cho, Kamigyo-ku, Kyoto 602-8566, Japan; aki-y@koto.kpu-m.ac.jp (A.Y.); ku-n07@koto.kpu-m.ac.jp (N.O.); c1981311@koto.kpu-m.ac.jp (Y.C.); koma@koto.kpu-m.ac.jp (N.T.); kaneko-y@koto.kpu-m.ac.jp (Y.K.); uchino@koto.kpu-m.ac.jp (J.U.); takayama@koto.kpu-m.ac.jp (K.T.); 2Department of Respiratory Medicine Uji-Tokushukai Medical Center, Kyoto 611-0041, Japan; dyckw344@yahoo.co.jp; 3Department of Respiratory Medicine, Japanese Red Cross Kyoto Daini Hospital, Kyoto 602-8026, Japan; k-hirose09@outlook.jp (K.H.); ytkkbt@yahoo.co.jp (Y.K.); 4Department of Respiratory Medicine, Japanese Red Cross Kyoto Daiichi Hospital, Kyoto 605-0981, Japan; sshiotsu@gmail.com; 5Department of Respiratory Medicine, Otsu City Hospital, Otsu 520-0804, Japan; osamu319@true.ocn.ne.jp

**Keywords:** osimertinib, EGFR-T790M mutation, non-small cell lung cancer, biomarker, retrospective study

## Abstract

Osimertinib is a mutant-selective EGFR inhibitor that is effective against non-small cell lung cancer (NSCLC) in patients with the *EGFR*-T790M mutation, who are resistant to EGFR-tyrosine kinase inhibitors (EGFR-TKIs). However, the factors affecting response to osimertinib treatment are unknown. In this retrospective study, 27 NSCLC patients with the *EGFR*-T790M mutation were enrolled at five institutions in Japan. Among several parameters tested, the progression-free survival (PFS) associated with the initial EGFR-TKIs was positively correlated with the PFS after osimertinib treatment (*p* = 0.021). The median PFS following osimertinib treatment and the overall survival (OS) were longer in patients who responded to osimertinib than in those who did not (17.7 months versus 3.5 months, *p* = 0.009 and 24.2 months versus 13.5 months, *p* = 0.021, respectively). A multivariate analysis demonstrated that the PFS with initial EGFR-TKIs was significantly related to the PFS with osimertinib treatment (*p* = 0.035), whereas osimertinib response was significantly related to the PFS and OS with osimertinib treatment (*p* = 0.016 and *p* = 0.006, respectively). Our retrospective observations indicate that PFS following the initial EGFR-TKI treatment and the response rate to osimertinib might be promising predictors for effective osimertinib treatment in NSCLC patients with the *EGFR*-T790M mutation.

## 1. Introduction

The development of molecular-targeted therapy has markedly improved clinical outcomes in non-small cell lung cancer (NSCLC) patients with alterations in the driver genes. NSCLC patients with activating epidermal growth factor receptor (EGFR) mutations, such as exon 19 deletion and exon 21 point mutation (L858R), respond significantly better to first- and second-generation EGFR-tyrosine kinase inhibitors (EGFR-TKIs) than to platinum-based chemotherapy [1,2]. However, almost all patients acquire resistance to initial EGFR-TKIs in approximately 10–12 months. Several acquired resistance mechanisms to initial EGFR-TKIs are known, including gatekeeper mutations such as *EGFR*-T790M, activation of bypass signaling, epithelial mesenchymal transition and transformation to small-cell lung cancer [3,4]. The *EGFR*-T790M mutation is the most common acquired resistance mechanism to first- and second-generation EGFR-TKIs [3,5]. Phase III clinical trials have revealed that the third-generation EGFR-TKI osimertinib provides a better progression-free survival (PFS) than platinum-based chemotherapy in NSCLC patients who have *EGFR*-T790M [6]. Therefore, osimertinib has been approved in the United States, Japan, and other countries for cases with EGFR-mutated NSCLC harboring EGFR-T790M mutations and acquired resistance to initial EGFR-TKIs, including gefitinib, erlotinib, and afatinib. More recently, osimertinib was approved as the first-line of treatment for advanced *EGFR*-mutated NSCLC patients based on the results of a phase III clinical trial [7].

Although osimertinib is effective in most NSCLC patients with the *EGFR*-T790M mutation, a 71% objective response rate and 10.1 months median PFS were recorded in a clinical trial; however, the mechanisms underlying patient response to osimertinib treatment are still unclear [6]. Ariyasu et al. reported that an increased expression of the T790M allele product is among the several EGFR-activating mutations that have been predicted to be involved in the response to osimertinib using droplet digital polymerase chain reaction analyses [8]. However, further investigations are warranted to reveal biomarkers that can more conveniently and cost-effectively detect responders to osimertinib among NSCLC patients with *EGFR*-T790M mutations because 6% of them showed the disease progression treated with osimertinib in a clinical trial [6].

In this retrospective study, we investigated predictive clinical biomarkers associated with osimertinib efficacy based on the profiles of NSCLC patients with the *EGFR*-T790M mutation after acquiring initial resistance to EGFR-TKIs.

## 2. Results

### 2.1. Patient Characteristics

A total of 78 EGFR-mutant NSCLC patients underwent re-biopsy to detect *EGFR*-T790M mutation after acquiring resistance to the initial EGFR-TKIs. After excluding 51 patients who met the exclusion criteria, 27 patients with *EGFR*-T790M mutations were finally enrolled (Appendix A). Patient characteristics after detection of *EGFR*-T790M mutations are summarized in Table 1. The median age was 73 years (range, 44–84); 18 patients (66.7%) were female; 20 patients (74.1%) were nonsmokers; and the majority of patients (92.6%) indicated a Performance Status (PS) of 0 and 1. The most prevalent history of disease included the incidence of adenocarcinoma (96.3%), 10 patients (37.0%) had relapse after surgery, and all the patients had exon 19 deletion or an L858R deletion in exon 21; these were the most common mutation types in EGFR. Twenty-four patients (88.9%) responded to the initial EGFR-TKIs. *EGFR*-T790M mutation was detected in five patients (18.5%) through liquid biopsy, in whom tumor re-biopsy was not conducted. Our results showed all of the EGFR-T790M mutation was detected in tumors after acquiring resistance to EGFR-TKIs, but not in tumors at a baseline, which indicated that it was caused by acquired resistance.

### 2.2. Predictor Factors in Osimertinib Treatment

To evaluate the clinical factors to predict the efficacy of osimertinib treatment in NSCLC patients with *EGFR*-T790M mutations, we first categorized two groups: patients in whom the PFS with osimertinib treatment was more than 8 months were classified into the “long PFS group,” and patients in whom the duration was less than 8 months were classified into the “short PFS group”. Of the 27 patients with *EGFR*-T790M mutations, 17 patients (63.0%) belonged to the long PFS group and 10 patients (37.0%) belonged to the short PFS group. There was no significant difference in patient profiles of the two groups as shown in Table 2. 

We next examined the difference in treatment-related factors in both groups and the results are presented in Table 3. The response rate to osimertinib was significantly higher in patients from the long PFS group than in those from the short PFS group (88.2% versus 40.0%, *p* = 0.025). The rate of longer PFS with initial EGFR-TKIs (more than 8 months) tended to be higher in patients in the long PFS group than in those in the short PFS group (<8 months) (88.2% versus 50.0%, *p* = 0.065). Therefore, we focused on the two clinical parameters, PFS with initial EGFR-TKI treatment and the response rate to osimertinib, as treatment-related factors for osimertinib.

Of these parameters, median PFS with osimertinib was 17.7 months in the long PFS group (95% confidence interval [CI] 9.0–22.0 months) and was 3.2 months in the short PFS group (95% CI 1.2–9.6 months) (*p* = 0.021). Meanwhile, there was no significant difference in OS between the two groups (*p* = 0.337) (Figure 1A,B). Median PFS with osimertinib was 17.7 months in osimertinib responders (CR/PR) (95% CI: 9.0 months—not evaluable [NE]) and 3.5 months in osimertinib non-responders (SD/PD) (95% CI: 0.3 months—NE) (*p* = 0.009). In addition, the osimertinib responders had a longer OS than the non-responders (24.2 months [95% CI: 22.1 months—NE] and 13.5 months [95% CI: 0.3 months—NE], *p* = 0.021) although patients of the short PFS group showed a significantly better opportunity for undergoing platinum doublet therapy after acquiring resistance to osimertinib than patients with longer PFS with osimertinib (*p* = 0.041) (Figure 1C,D).

The multivariate analysis demonstrated that PFS with the initial EGFR-TKIs was significantly related to the PFS with osimertinib (HR 0.31, 95% CI = 0.11–0.92, *p* = 0.035), whereas osimertinib response was significantly related to the PFS with osimertinib and the OS (HR 0.29, 95% CI = 0.11–0.80, *p* = 0.016; HR 0.09, 95% CI = 0.02–0.50, *p* = 0.006, respectively) (Table 4). 

Given that the response rate with osimertinib was significantly correlated with the PFS, we further examined the response rate relative to the baseline according to exposure to osimertinib. The median maximum tumor shrinkage (MTS) of the group with longer PFS and short PFS were 50.0% and 20.6%, respectively, which indicated a significant association between PFS and osimertinib treatment (*p* = 0.006) (Figure 2A–C).

## 3. Discussion

The *EGFR*-T790M mutation was detected in approximately 50% of the patients, who had acquired resistance to first- and second-generation EGFR-TKIs [3,5]. We revealed that the response rate to initial EGFR-TKI administration positively correlates with the detection of T790M mutation in EGFR mutated NSCLC patients [9]. Moreover, this present study showed that initial EGFR-TKI response is a useful predictor for osimertinib treatment in NSCLC patients with the *EGFR*-T790M mutation. This finding is meaningful from the point of view of the detection of responders during initial EGFR-TKI treatment.

Osimertinib is known as a mutant-selective EGFR inhibitor that is effective for *EGFR*-T790M-positive NSCLC after acquired resistance to initial EGFR-TKIs [10,11]. There was a significant difference in the median PFS with osimertinib treatment between *EGFR*-T790M-positive and -negative NSCLC patients after acquiring resistance to initial EGFR-TKIs [12]. Wang S. et al. showed that patients with acquired *EGFR*-T790M mutation showed a higher frequency of exon 19 deletion in *EGFR*, shorter PFS with osimertinib, and prolonged OS than those with primary *EGFR*-T790M mutation [13]. In this study, all patients gained *EGFR*-T790M mutations from the acquired resistance phase, and there was no difference between 20 in exon 19 deletion in *EGFR* and 7 in exon 21 L858R mutation in *EGFR* with regard to clinical outcomes, such as PFS with osimertinib and OS.

However, such patients with *EGFR*-T790M-mutant NSCLC were expected to have other diverse resistance mechanisms that indicate the various responses to osimertinib because of the increase in intratumor heterogeneity during initial EGFR-TKI treatment. Indeed, the high tumor mutation burden in patients with EGFR-mutant NSCLC was involved in poor prognosis of EGFR-TKIs treatment [14]. Therefore, the dependency on EGFR signaling at the baseline might be critical for predicting osimertinib response after acquiring resistance to initial EGFR-TKIs in NSCLC patients with the *EGFR*-T790M mutation (Figure 2D).

Previous studies have shown that first-line EGFR-TKI responders have significantly longer PFS than the non-responders among EGFR mutant NSCLC patients [15]. The proportion of tumor cells that respond to osimertinib might be expected to correlate with the longer PFS associated with osimertinib treatment, in consistency with the initial EGFR-TKIs. Moreover, the population of *EGFR*-T790M mutation before osimertinib treatment might correlate with the dependency on EGFR signaling, which may lead to the prediction of the effectiveness of the osimertinib treatment (Figure 2D).

Several mechanisms were reported for the acquired resistance to osimertinib in *EGFR*-T790M mutated NSCLC patients, including *EGFR*-C797S mutation, bypass signal activation, and transformation to small cell lung cancer [16,17,18]. To overcome these resistance mechanisms, combined therapies, such as combination with anti-angiogenesis inhibitors, are ongoing in multiple clinical trials. In addition, preclinical studies have showed that combination with osimertinib may be promising for *EGFR*-mutant NSCLC patients with or without T790M mutations [19,20]. The NSCLC patients with *EGFR*-T790M mutation after acquiring resistance to osimertinib had significantly longer PFS in osimertinib treatment than those without *EGFR*-T790M mutation [21]. Interestingly, patients that did not have *EGFR*-T790M mutation induced EGFR-independent mechanisms on acquiring resistance, such as activation of bypass signaling and transformation to small cell lung cancer. Our observations indicate a positive correlation between the response to osimertinib and the PFS with osimertinib treatment. These findings suggest that the population of EGFR-dependent tumor cells at the time of pretreatment with osimertinib may have an impact on the PFS following osimertinib treatment (Figure 2D).

The emergence of *EGFR*-T790M mutation is reported as a good prognosis factor, when the patient acquires resistance to initial EGFR-TKIs [22]. A previous study demonstrated that high expression of AXL in pre-treatment tumors was relatively related to poor outcomes of EGFR-TKI treatment, including osimertinib [20]. Our observations also revealed that the response rate of osimertinib has a significant relationship with survival. Therefore, patients with longer PFS with osimertinib treatment are expected to maintain the tumors with *EGFR*-T790M mutation after acquiring resistance to osimertinib; this observation suggests that PFS might be involved in survival. However, further studies are required to validate this point.

This study has several limitations, which are as follows: First, it has a small sample size and was retrospective in nature. However, the median PFS of all the patients in this study was similar to that observed in a phase III study (9.6 months [95% CI: 5.1–19.7] vs. 10.1 months [95% CI: 8.3–12.3]) [6]. Second, this study was based on a Japanese cohort of patients only. Third, we had various biases on patient conditions when EGFR-TKIs were started, even though the study was performed in multiple centers and the timing of evaluation by CT scanning was in the following 1–3 months. Therefore, further prospective study is warranted to identify the role of longer PFS duration of the initial EGFR-TKIs administration on the osimertinib response in *EGFR*-T790M-positive NSCLC.

## 4. Materials and Methods

### 4.1. Patients

We retrospectively enrolled 39 patients with advanced or postoperative recurrent EGFR-mutant NSCLC; re-biopsy samples were obtained from these patients. The samples were either from tumors or the plasma after resistance was acquired to the initial EGFR-TKIs. The patients were enrolled at five institutions in Japan between May 2014 and January 2018. The exclusion criteria were as follows: (1) no osimertinib treatment, (2) discontinuation of initial EGFR-TKI treatment owing to adverse events, and (3) no measurable lesions.

All patients were evaluated for imaging responses, including complete response (CR), partial response (PR), stable disease (SD), and progressive disease (PD), by conventional CT scanning according to the instructions of RECIST version 1.1. We obtained patients’ clinical data from medical records retrospectively; the information included age, sex, smoking status, Eastern Cooperative Oncology Group Performance Status (PS), histological subtype, clinical stage, EGFR mutation status, initial EGFR-TKI administered, initial EGFR-TKI response, PFS with initial EGFR-TKI, re-biopsy site, history of platinum-based chemotherapy after acquiring resistance to osimertinib, PFS with osimertinib, and overall survival (OS). We set the initial EGFR-TKI PFS cutoff to eight months because the median PFS with gefitinib and erlotinib were reported as approximately eight months in phase 3 clinical trial [23]. PFS was defined as the period from osimertinib treatment initiation to disease progression by RECIST or the period till 30 June, 2018. OS was defined as the period from osimertinib treatment initiation to death or until 30 June, 2018. This study protocol was approved by the Ethics Committees (Kyoto Prefectural University of Medicine) of each hospital on 08 February 2018 (ethic code: ERB-C-1107). The TNM stage was classified using version 7 of the TNM stage classification system.

### 4.2. EGFR Mutation Analysis

EGFR mutations were detected using the polymerase chain reaction method for tumor and plasma samples by sequencing exons 18–21; the sequencing was performed at commercial clinical laboratories: SRL, Inc. and BML, Inc. (Tokyo, Japan).

### 4.3. Statistical Analysis

The Cox proportional-hazard model, which accounted for several factors of the patient profiles was used. To analyze the PFS, times to events were estimated using the Kaplan–Meier method and compared by the log-rank test. The PFS was censored at the date of disease progression. Predictive factors for osimertinib response to *EGFR*-T790M mutation in NSCLC patients were identified using univariate and multivariate logistic analyses. All statistical analyses were performed using EZR for Windows, version 1.35 (Saitama Medical Center, Jichi Medical University, Saitama, Japan). *p* values less than 0.05 indicated statistical significance.

## 5. Conclusions

Our retrospective observations suggest that the PFS with initial EGFR-TKI treatment and the response rate to osimertinib might be promising predictors for osimertinib treatment in patients with *EGFR*-T790M-positive NSCLC; this may be due to the ratio of tumor heterogeneity that might be enriched during initial EGFR-TKI treatment. Further experiments are needed to validate these observations.

## Figures and Tables

**Figure 1 cancers-11-00365-f001:**
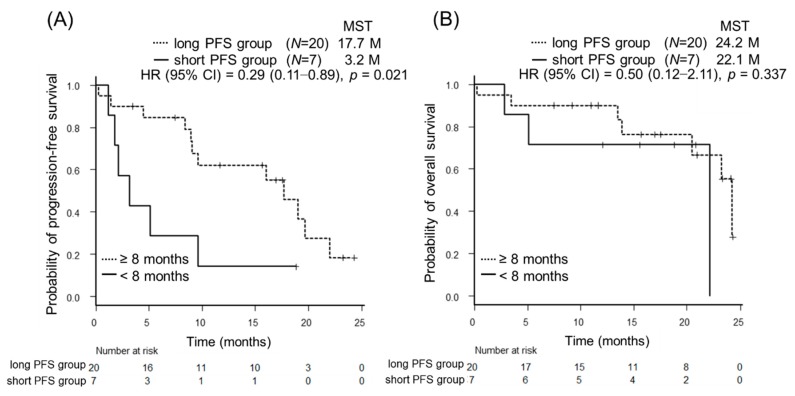
Kaplan-Meier survival curves for progression free survival (PFS) and overall survival (OS) for the PFS duration of initial EGFR-TKI treatment and osimertinib response. (**A**,**B**) PFS and OS of the *EGFR*-T790M-mutant non-small cell lung cancer (NSCLC) patients with the long PFS for the initial EGFR-TKI (*N* = 20) or those with the short PFS (*N* = 7). The median PFS was significantly longer in patients with the long PFS for the initial EGFR-TKI than in those with the short PFS (17.7 months versus 3.2 months, *p* = 0.021). There was no significant difference in OS between the two groups (*p* = 0.337). (**C**,**D**) PFS and OS of *EGFR*-T790M-mutant NSCLC patients with osimertinib responsiveness (CR/PR) (*N* = 19) or those with osimertinib non-responsiveness (SD/PD) (*N* = 8). The median PFS and OS were significantly longer in osimertinib responders than in osimertinib non-responders (17.7 months versus 3.5 months, *p* = 0.009), (24.2 months versus 13.5 months, *p* = 0.021), respectively.

**Figure 2 cancers-11-00365-f002:**
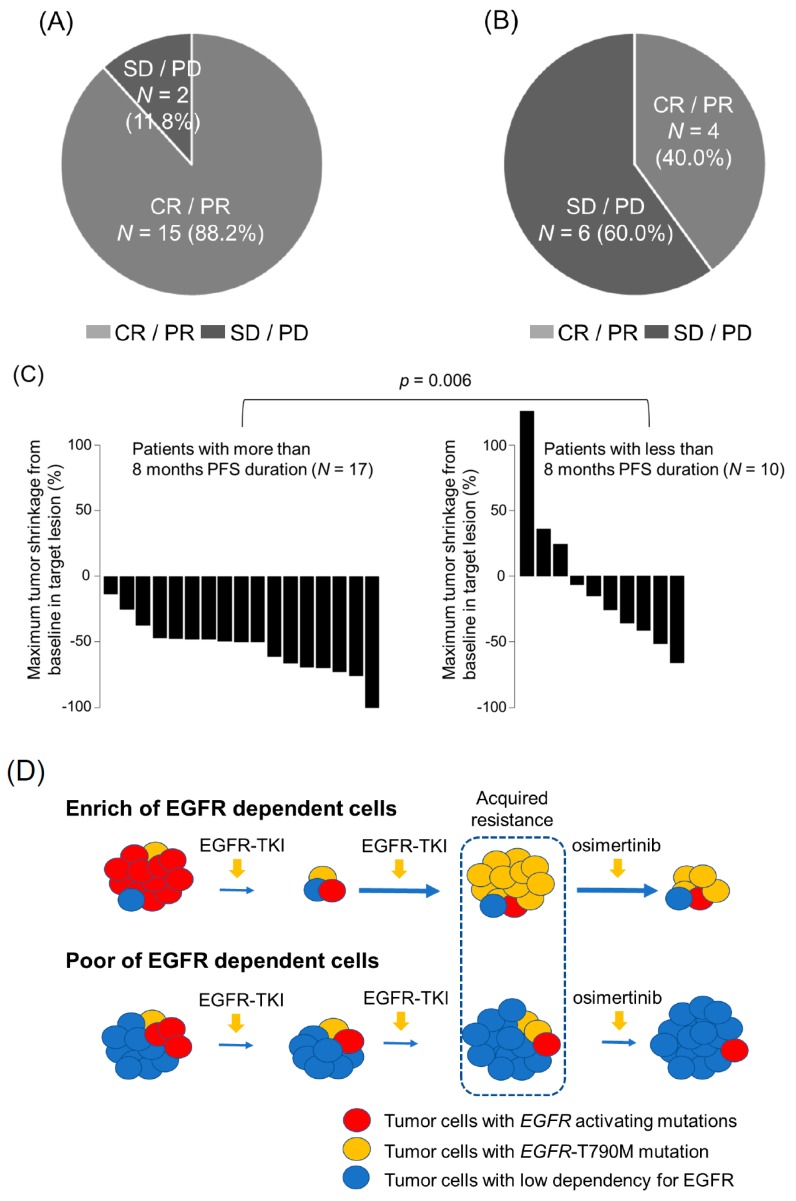
The osimertinib response of NSCLC patients with *EGFR*-T790M mutation treated with osimertinib on the PFS duration of osimertinib. (**A**) Frequency of the best overall response to osimertinib treatment among *EGFR*-T790M mutant NSCLC patients with the more than 8 months PFS duration (*N* = 17). (**B**) Frequency of the best overall response to osimertinib treatment among *EGFR*-T790M-mutant NSCLC patients with the less than 8 months PFS duration (*N* = 10). (**C**) The median maximum tumor shrinkage rate relative to baseline in 27 NSCLC patients with *EGFR*-T790M mutation treated with osimertinib. The PFS duration of 17 patients was more than 8 months and that of 10 patients was less than 8 months. The median maximum tumor shrinkage rate in these patients indicated a significant association with the osimertinib response (50.0% and 20.6%, respectively, *p* = 0.006). (**D**) Schematic diagram showing that tumors with resistance to EGFR-TKIs may be heterogeneous, consisting of both EGFR signal dependency with *EGFR* activating mutation (red) or with *EGFR*-T790M mutation (yellow), and EGFR signal independency (blue) populations.

**Table 1 cancers-11-00365-t001:** Patients baseline characteristics.

Patients’ Characteristics		*N* = 27, *n*, (%)
Age	Median (Range)	78.0 (47.0–88.0)
Sex	Male	9 (33.3)
	Female	18 (66.7)
PS	0, 1	25 (92.6)
	2	2 (7.4)
Histology	Adenocarcinoma	26 (96.3)
	Squamous cell carcinoma	1 (3.7)
Smoking status	Never-smoker	20 (74.1)
	Ever-smoker	4 (14.8)
	Current-smoker	3 (11.1)
Stage	III	4 (14.8)
	IV	13 (48.1)
	Postoperative recurrence	10 (37.0)
EGFR mutation status	Exon 19 deletion	20 (74.1)
	Exon 21 L858R mutation	7 (25.9)
Initial EGFR-TKI	Gefitinib	17 (63.0)
	Erlotinib	6 (22.2)
	Afatinib	4 (14.8)
Initial TKI response	CR, PR	24 (88.9)
	SD, PD	3 (11.1)

Abbreviations: PS, performance status; EGFR, epidermal growth factor receptor; EGFR-TKI, EGFR-tyrosine kinase inhibitor; TKI, tyrosine kinase inhibitor; CR, complete response; PR, pertial response; SD, stable diseaase; PD, progressive disease; *N*, number.

**Table 2 cancers-11-00365-t002:** Patients baseline characteristics classified by progression-free survival (PFS) duration of osimertinib.

Patients’ Characteristics		PFS Duration of Osimertinib ≥ 8 Months	PFS Duration of Osimertinib < 8 Months	*p* Value
		*N* = 17	*N* = 10	
		*n* (%)	*n* (%)	
Age	Median (Range)	78.0 (49.0–88.0)	71.0 (47.0–83.0)	0.247
Sex	Male	5 (29.4)	4 (40.0)	0.683
	Female	12 (70.6)	6 (60.0)	
PS	0, 1	16 (94.1)	9 (90.0)	1
	2	1 (5.9)	1 (10.0)	
Histology	Adenocarcinoma	17 (100.0)	9 (90.0)	0.37
	Squamous cell carcinoma	0 (0.0)	1 (10.0)	
Smoking status	Never-smoker	12 (70.6)	8 (80.0)	0.678
	Smoker	5 (29.4)	2 (20.0)	
Stage	III	2 (11.8)	2 (20.0)	0.472
	IV	7 (41.2)	6 (60.0)	
	Postoperative recurrence	8 (47.1)	2 (20.0)	
EGFR mutation status	Exon 19 deletion	12 (70.6)	8 (80.0)	0.678
	Exon 21 L858R mutation	5 (29.4)	2 (20.0)	
EGFR-TKI	Gefitinib	11 (64.7)	6 (60.0)	0.195
	Erlotinib	5 (29.4)	1 (10.0)	
	Afatinib	1 (5.9)	3 (30.0)	
Re-biopsy site	Intrathoracic	12 (70.6)	5 (50.0)	0.623
	Extrathoracic	2 (11.8)	3 (30.0)	
	Liquid	3 (17.6)	2 (20.0)	

**Table 3 cancers-11-00365-t003:** Patients’ characteristics of clinical course classified progression-free survival duration of osimertinib.

Patients’ Characteristics		PFS Duration of Osimertinib ≥ 8 Months	PFS Duration of Osimertinib < 8 Months	*p* Value
		*N* =17	*N* = 10	
		*n* (%)	*n* (%)	
PFS duration of initial TKI	≥8 months	15 (88.2)	8 (80.0)	0.065
	<8 months	2 (11.8)	2 (20.0)	
Osimertinib response	CR/PR	15 (88.2)	4 (40.0)	0.025
	SD/PD	2 (11.8)	6 (60.0)	
Osimertinib shrinkage	Median (Range)	50.0 (13.5–100.0)	20.6 (−126.4–66.5)	0.006
Platinum doublet therapy after osimertinib	+	0 (0.0)	3 (30.0)	0.041
	-	17 (100.0)	7 (70.0)	

**Table 4 cancers-11-00365-t004:** Univariate and multivariate analysis of patients’ characteristics and the clinical course.

Variables	Progression-Free Survival				Overall Survival			
	Univariate Analysis		Multivariate Analysis		Univariate Analysis		Multivariate Analysis	
	HR, Mean (95% CI)	*p* Value	HR, Mean (95% CI)	*p* Value	HR, Mean (95% CI)	*p* Value	HR, Mean (95% CI)	*p* Value
Age at detection of T790M (<75/≥75 years)	1.38 (0.53–3.57)	0.504			0.71 (0.20–2.55)	0.603		
Performance status (0–1/2)	0.27 (0.06–1.26)	0.073			NE	0.557		
Disease stage (postoperative recurrence/ stage III or IV)	0.31 (0.1–0.95)	0.03	0.41 (0.12–1.41)	0.16	0.30 (0.06–1.41)	0.105		
EGFR status (exon19 deletion/exon21 L858R mutation)	0.84 (0.45–1.58)	0.177			1.53 (0.31–7.42)	0.597		
EGFR-TKI (afatinib/gefitinib, erlotinib)	3.01 (0.81–11.16)	0.084			4.05 (0.96–17)	0.039	8.15 (1.10–60.38)	0.04
Initial EGFR-TKI response (CR, PR/SD, PD)	0.63 (0.14–2.83)	0.543			4.89 (0.61–39.16)	0.099		
Initial EGFR-TKI PFS (more than 8 months/less than 8 months)	0.31 (0.11–0.89)	0.021	0.30 (0.10–0.90)	0.031	0.50 (0.12–2.11)	0.337		
Re-biopsy (tissue/liquid)	0.62 (0.17–2.31)	0.476			NE	0.319		
Osimetinib response (CR, PR/SD, PD)	0.29 (0.11–0.78)	0.009	0.44 (0.15–1.33)	0.15	0.09 (0.02–0.47)	<0.001	0.02 (0.00–0.27)	0.002

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
