# Peer review of "Clinical Characteristics of Osimertinib Responder in Non-Small Cell Lung Cancer Patients with EGFR-T790M Mutation"

_cancers, 2019, doi:10.3390/cancers11030365_

Reviewer 1 Report

This study provides a valuable clinical evidence to predict the efficacy of osimertinib, and clarifies that osimertinib is potential to overcome drug resistance in response to the first- and second-generation TKIs. The authors also showed excellent statistical analysis to compare different groups. This study will receive abundant attentions if sample size continues to be increased. 

One suggestion

The limitation of osimertinib deserves to be described with more detail information in Introduction.

Some minor questions:

Why 8-months was set as a "cutoff" ??

Whether combination of osimertinib with the other TKI is work for NSCLC ?? Authors may provide opinions and some references in discussion.

In addition to evidence provided by this manuscript, is there any other factor potential to predict efficacy of osimertinib ?? (Discussion)

Author Response

Reviewer1

One suggestion

The limitation of osimertinib deserves to be described with more detail information in Introduction.

Reply 

As the reviewer1 recommended, we added the sentence regarding the limitation of osimertinib for the treatment of NSCLC with T790M mutation in the introduction sections (page 2, lines 52, 58-59).

Some minor questions:

Why 8-months was set as a "cutoff" ??

Reply 

We added the sentence regarding “cutoff” in method section (page 9, lines 220-222).; as followed: “We set the initial EGFR-TKI PFS cutoff to eight months because the median PFS with gefitinib and erlotinib were reported approximately eight months in phase 3 clinical trial.”

Whether combination of osimertinib with the other TKI is work for NSCLC ?? Authors may provide opinions and some references in discussion.

Reply 

To the best our knowledge, the combined therapy with osimertinib has not been approved for EGFR-mutant NSCLC patients. This concept is ongoing to conduct multiple clinical trials, such as combination with anti-angiogenesis inhibitors. In addition, several study reported the combination of osimertinib is promising for EGFR-mutant NSCLC patients with or without T790M mutations. We added these sentences in the discussion part, as followed; “To overcome these resistance mechanisms, the combined therapies are ongoing by multiple clinical trials, such as combination with anti-angiogenesis inhibitors. In addition, preclinical study showed the combination with osimertinib may be promising for EGFR-mutant NSCLC patients with or without T790M mutations.” (page 8, lines 177-180).

In addition to evidence provided by this manuscript, is there any other factor potential to predict efficacy of osimertinib ?? (Discussion)

(Reply) 

Our previous study showed that AXL high expression in the pre-treatment tumors was relatively related to poor outcomes of EGFR-TKI treatment, including osimertinib. We added these results in the discussion part (page 9, lines 190-192).

Reviewer 2 Report

In the study entitled "Clinical characteristics of osimertinib responder in non-small cell lung cancer patients with EGFR-T790M mutation" the authors investigated predictive clinical biomarkers associated with osimertinib efficacy on the profiles of NSCLC patients with the EGFR-T790M mutations.

By using a multivariate analysis, they demonstrate that the PFS with initial EGFR-TKIs was significantly related to the PFS with osimertinib treatment. Therefore,tis might be a promising predictor for effective osimertinib treatment in NSCLC patients with  EGFR-T790 mutations.

Although the topic is very interesting and the study may have a relevant implication in the correct use of osimertinib in clinical trials , there are some points that need to be addressed.

Major Points

The Authors do not report if EGFR-T790M mutation is a primary or acquired mutation. Do they know some informations about this important baseline condition? If yes, please insert and comment properly this new information.

Recently, it has been report a retrospective study comparing the efficacy of osimertinib in patients with primary and acquired T790M mutations, by exploring the differences in clinical and molecular charatcteristics (Wang, S. , Yan, B. , Zhang, Y. , Xu, J. , Qiao, R. , Dong, Y. , Zhang, B. , Zhao, Y. , Zhang, L. , Qian, J. , Lu, J. , Zhao, R. and Han, B. (2019), Different characteristics and survival in non‐small cell lung cancer patients with primary and acquired EGFR T790M mutation. Int. J. Cancer. doi:10.1002/ijc.32015). They analyzed the response to first generation TKIs in both groups (for example PFS and OS ). Therefore it would be usefull to comment the data obatined in this paper with this new data, to better  help clinical to choose the correct therapy.

Minor Points

In Fig.2c  the authors reported the same legends (Patients with more than...)

Author Response

Reviewer2

Major Points

The Authors do not report if EGFR-T790M mutation is a primary or acquired mutation. Do they know some informations about this important baseline condition? If yes, please insert and comment properly this new information.

(Reply) 

We appreciate the reviewer2 for these precious suggestions. Our results showed the all of EGFR-T790M mutation detected in tumors after acquired resistance to EGFR-TKIs, but not in tumors at a baseline, indicated that it was caused by acquired resistance. As the reviewer recommended, we added the information into the result sections (page 2, lines 75-77).

Recently, it has been report a retrospective study comparing the efficacy of osimertinib in patients with primary and acquired T790M mutations, by exploring the differences in clinical and molecular characteristics (Wang, S. , Yan, B. , Zhang, Y. , Xu, J. , Qiao, R. , Dong, Y. , Zhang, B. , Zhao, Y. , Zhang, L. , Qian, J. , Lu, J. , Zhao, R. and Han, B. (2019), Different characteristics and survival in non‐small cell lung cancer patients with primary and acquired EGFR T790M mutation. Int. J. Cancer. doi:10.1002/ijc.32015). They analyzed the response to first generation TKIs in both groups (for example PFS and OS). Therefore, it would be useful to comment the data obtained in this paper with this new data, to better help clinical to choose the correct therapy.

 (Reply) 

We appreciate for providing the information of curious retrospective study on our research. In this study, EGFR-T790M mutation was detected in tumors after the acquired resistance but not in the initial phase. Wang S, et al. showed that patients with acquired EGFR-T790M mutation was detected in more frequency of exon 19 deletion in EGFR, shorter PFS with osimertinib, and prolonger OS than that with primary EGFR-T790M mutation. In this study, all patients gained EGFR-T790M mutations for acquired resistance phase, and were not the difference between 20 in exon 19 deletion in EGFR and 7 in exon 21 L858R mutation in EGFR regarding clinical outcomes, such as PFS with osimertinib and OS. However, as the reviewer pointed, the research by Wang S, et al. was helpful to understand the clinical feature of patients with T790M mutation. We added this results in the discussion part (page 8, lines 156-161).

Minor Points

In Fig.2c  the authors reported the same legends (Patients with more than...)

 (Reply) We appreciate for the reviewer’s points. We revised these parts the reviewer pointed.

Round  2

Reviewer 2 Report

After re-reading the manuscript, I think the authors have tried to address most of comments, giving a significant improvements in the manuscript.